

# Opposing changes in subpolar ocean heat content due to meridional heat advection driven by the Southern Ocean wind anomaly

XuBin Ni[1,2], Ling Du[1,2], HuangYuan Shi[3]

[1]Frontier Science Center for Deep Ocean Multispheres and Earth System (FDOMES) and Physical Oceanography
Laboratory, Ocean University of China, Qingdao, 266100, China
[2]College of Oceanic and Atmospheric Sciences, Ocean University of China, Qingdao, 266100, China
[3] College of Ocean and Meteorology, Guangdong Ocean University, Zhanjiang, 524000, China

*Correspondence to*: Ling Du (duling@ouc.edu.cn)

**Abstract.** The global ocean has been warming significantly due to rapid climate change, leading to conspicuous changes in the subpolar Southern Ocean. Our study reveals that the heat exchange between Antarctic and subtropical oceans driven by wind, which plays an important role in modulating changes in regional ocean heat content (OHC) through meridional heat advections. In this study, we used the observed objective analysis and reanalysis datasets to explore the changes in subpolar ocean heat content and analyze attributions to the remarkable regional discrepancy. We found a notable difference in OHC
trends between the Atlantic–Indian sector and the Pacific sector, which could be attributed to the inverse meridional heat advection caused by wind anomalies. Atlantic–Indian sector warming was significantly modulated by increasing meridional heat advection induced by the poleward westerly wind. In the Pacific sector, the enhanced wind resulted in substantial cold-water advection equatorward, causing significant cooling. These opposite advections are also occurring with the corresponding regional front movement, which also indicates the meridional heat exchange between oceans. This study
highlights that wind anomalies play an important role in modulating the heat exchange between Antarctic and subtropical oceans. Consequently, the atmospheric forcing may become more significant to the heat redistribution in the Southern Ocean in the warmer future.

## 1 Introduction

The Southern Ocean has experienced widespread warming in response to climate change for several decades (Sallée, 2018;
Liu et al., 2018; Jones et al., 2019; Newsom et al., 2021; Storto et al., 2021). This significant warming manifests in an increase in ocean heat content (OHC), which is considered an effective climate change proxy (Storto et al., 2021). Many studies have recorded remarkable increases in OHC in the Southern Ocean (Kolodziejczyk et al., 2019; Carter et al., 2022). The Special Report on the Ocean and Cryosphere in a Changing Climate, published by the Intergovernmental Panel on Climate Change (IPCC), highlights the critical role of the Southern Ocean in global ocean heat gain where it was determined
that the upper-2000 m of the Southern Ocean accounted for 35–43% of the global ocean heat gain with high confidence



(Meredith et al., 2019). From observations and simulations, there is substantial anthropogenic heat induced by poleward-intensified wind changes and this massive heat is also mostly absorbed in the Southern Ocean (Liu et al., 2018) during the past few decades particularly in the subpolar ventilated regions (south of 50°S) upper 2000 m depths (Sallée, 2018; Shi et al., 2018). However, subpolar warming is heterogeneous and exhibits significant regional changes (Sallée, 2018). For example,
the Atlantic and Indian sectors where warming is particularly pronounced (Shi et al., 2018; Yang et al., 2020). In contrast, recent studies (e.g. Chung et al., 2022; Kang et al., 2023) have observed OHC decreasing in the Pacific sector, indicating opposing OHC changes between different basins.

Recently, persistent warming has occurred at all depths in the Atlantic and Indian sectors and at the same time, significant surface cooling in the Pacific sector has attracted much attention (Kolodziejczyk et al., 2019; Wang et al., 2021; Chung et al.,
2022; Kang et al., 2023). These remarkable anomalies are closely related to the regional heat transport/advection, which is an important part of the redistribution component (Zika et al., 2021). With the development and support of the Coupled Model Intercomparison Project Phase 6 (CMIP6), OHC changes can be attributed to two main factors: heat uptake due to anthropogenic influence (e.g., Liu et al., 2018; McBride et al., 2021) and heat redistribution (e.g., Clément et al., 2022; Silvy et al., 2022) associated with heat transport/advection (Silvy et al., 2022). The redistribution component covers considerable
heat changes due to important horizontal heat advection (Zika et al., 2021). In the Southern Ocean, the Subantarctic Front (SAF) acts as a strong barrier. Heat advection, and particularly meridional heat advection, plays a critical role in OHC variations, bringing heat across the strong barrier.

Westerly wind is an important driver of Antarctic and Southern Ocean Climate and usually act on the meridional heat transport in the atmosphere as well as upper ocean (Meredith et al., 2019). The strong zonal wind mainly drives ocean heat
changes via Ekman pumping and Ekman transport over the upper Southern Ocean. Observations demonstrate that upper ocean gains more heat by the increasing subduction and thickness of the Subantarctic Mode Water in the south Indian ocean since robust westerly wind enhances the wind stress curl, which facilitates stronger downward Ekman pumping (Gao et al., 2018; Qu et al., 2020; Portela et al., 2020). At the same time, increasing westerly wind also promotes the upwelling of the warm circumpolar deep water and consequently the intermediate ocean warms significantly (Martinson et al., 2012; Herraiz-
Borreguero and Naveira Garabato et al., 2022). Strengthened westerly wind also drives stronger Ekman transport, which is an important seasonal part of the meridional ocean heat transport changes in the upper Southern Ocean (Yang and Saenko, 2018; Waugh et al., 2019). The increasing northward Ekman transport also brings about sufficient cold water and induced surface cooling near the Antarctic (Armour et al., 2016), especially in the South Pacific. The heat transport driven by wind anomalies induces regional heat exchange, which not only contributes to the regional OHC changes but also enable the
movement of fronts between Antarctic and subtropical oceans. And the movement implies the upper limb of the Southern Ocean meridional overturning circulation changes in regions.

The regional changes in OHC contain the heat transport in the ocean (e.g. Clément et al., 2022; Silvy et al., 2022). Indeed, heat advection is an important part of heat transport, and both describe ocean heat redistribution. However, heat advections emphasize the influence of temperature gradients as well as the regional current changes. Therefore, heat advection plays an



important role in modulating regional ocean thermal variations considering the strong nearby meridional temperature gradient as well as stronger currents in the subpolar Southern Ocean. That is, the changes of meridional current as well as meridional temperature gradient can reflect variations in meridional heat advection, which can better capture the regional heat exchange between Antarctic and subtropical oceans. The meridional heat and cold-water advection occurring in the Southern Ocean where there is conspicuous tilting of isopycnals, implying that there is also significant temperature and

salinity changes along the isopycnal. Hence, we applied heave and spice decomposition on meridional heat advection effects on heat variations. Then, OHC variations due to the meridional heat advection include two components: the heave and spice, as described by (Zunino et al., 2012). Heave represents the vertical displacements of the isopycnals and reflects the influence of advection on the displacement of isopycnals, and it can effectively capture increases in OHC in the Southern Ocean (Roberts et al., 2017; Desbruyères et al., 2017). While spice refers to the changes along isopycnals (e.g., Wang et al., 2021;

Chen and Cheng, 2023) and describes temperature changes along isopycnals through advection in the upper ocean, particularly in the ventilated regions of the subpolar Southern Ocean, with intense vertical mixing (Desbruyères et al., 2017). Heave and spice also perform well in manifesting the temperature anomalies within the ocean interior (Clément et al., 2022). Importantly, the pattern of subpolar OHC changes exhibits a remarkable discrepancy, but the fundamental factors influencing this discrepancy are unclear. We emphasize the meridional heat advection driven by wind anomalies, which

modulates heat exchange between Antarctic and subtropical oceans and this has been overlooked in previous studies. This is beneficial for understanding the contribution of redistributions to characteristics of regional heat content. The study is organized as follows: Section 2 provides the adopted dataset and methods. The variations of recent OHC and regional heat advection driven by wind anomaly are investigated in Section 3. Section 4 summarize the results and provide further discussion.

**2 Data and methods**

**2.1 Data**

In the present study, we employed four ocean temperature and salinity datasets to analyze the heat content characteristics in the subpolar sectors of the Southern Ocean. An objective analysis dataset, Ishii version 7.3, provided by Ishii et al. (2017) was based on Argo and other ocean measurements which were updated to March 2023. This monthly dataset has a $1 \times 1°$

horizontal resolution and 28 levels from 1955 to 2019. The IPCC Special Report on the Ocean and Cryosphere in a Changing Climate has adopted OHC trends derived from Ishii and other datasets and these trends were consistent with the Southern Ocean (Meredith et al., 2019). A recent study by Yang et al. (2020) utilizes the Ishii dataset to analyze OHC variations in the Southern Ocean. The Ishii dataset also performs well in investigating the temperature changes in the heave-spice components (Wang et al., 2021) and in presenting discrepancies in basin-scale OHC changes (Wang et al., 2018). In

this study, the results of OHC variations and water mass property changes were primarily based on the Ishii dataset.



In addition to the Ishii dataset, three other ocean temperature/salinity datasets used were as follows. Observed grid data based on Argo floats and the Grid Point Value of the Monthly Objective Analysis (MOAA GPV) (Hosoda, 2007) were provided by the Japan Agency for Marine-Earth Science and Technology (JAMSTEC). MOAA GPV consists of monthly objective analysis fields at a resolution of $1 \times 1°$ with 25 depth levels, spanning from January 2001 to March 2023. The World Ocean Atlas 2018 (WOA18) (Locarnini et al., 2018; Zweng et al., 2019) was also utilized, which provided objectively analyzed climatological mean fields at a horizontal resolution of $1 \times 1°$ with 102 depth levels from 0 to 5500 m. The periods of 1975–1984, 1985–1994, 1995–2004, and 2005–2017 from WOA18 were adopted in our study. Pre-1993 OHC was based on the former two periods and post-2005 OHC was based on the latter two periods. The observed temperature/salinity profiles were obtained from the World Ocean Database 2018 (WOD 18), which is a quality-controlled dataset provided by the National Centers for Environmental Information (NCEI) (Boyer et al., 2018). The widely used observation, Hydrographic Programme of the international World Ocean Circulation Experiment (WOCE) is also adopted in this paper and it is available in CLIVAR and Carbon Hydrographic Data Office (CCHDO). We selected five sections from WOCE near the study regions and more information about used sections is shown the supplement (Supplementary Table 1). We compared the temperature/salinity profiles in the Ishii dataset with the objective and observed datasets to better demonstrate and understand variations in OHC (Supplementary Figures 2, 3, 4).

The Simple Ocean Data Assimilation version 3.4.2, (SODA 3.4.2) ocean reanalysis (Carton et al., 2019) was obtained from the Department of Atmospheric and Oceanic Science, University of Maryland. Monthly current velocity is adopted in this study, with a $0.5 \times 0.5°$ resolution and 50 levels from 1980 to 2017. The ocean currents in version 3.4.2 behave well, as it improves eddy-permitting spatial resolution, active sea ice, and bias adjustment.

The wind fields in 10m were produced by the European Centre for Medium-Range Weather Forecasts (ECMWF) Integrated Forecast System (IFS) and derived from the fifth generation ECMWF atmospheric reanalysis (ERA5). ERA5 is freely available from the Copernicus Climate Change Service (C3S) in Climate Data Store (CDS). ERA5 has combined model data with observations and provided atmospheric variables with longer term from 1979 and then has taken the place of ERA-Interim reanalysis. The horizontal resolution of monthly wind fields is 0.25°×0.25° (Hersbach et al., 2019).

The sea ice concentration (SIC) was derived from passive microwave data that was achieved by satellites and obtained from the American National Snow & Ice Data Center (NSIDC). The NOAA/NSIDC Climate Data Record of Passive Microwave Sea Ice Concentration Version 3 has a $25 \times 25$ km horizontal resolution (Comiso and Nishio, 2008). In this study, monthly SICs were employed to support the surface cooling by the cold-water advection.

## 2.2 Method

This study pays close attention to the OHC (unit: J) above 2000 m since this is where the OHC increases significantly, which was computed as follows (Kolodziejczyk et al., 2019):

$$OHC = \int_{z1}^{z2} \rho C_p T(z) dz \tag{1}$$



where $\rho$ is the seawater density and $C_p$ is the isobaric thermal capacity of the seawater, and T is the conservative temperature, which is computed by The Gibbs Sea Water (GSW) Oceanographic Toolbox of the Thermodynamic Equation of Seawater -

2010 (TEOS-10) (Roquet et al., 2015). The $z_1$ and $z_2$ parameters denote the upper and lower layers, respectively. To calculate OHC above 2000 m, z1 and z2 represent the upper and lower boundaries, respectively, where $z_1 = 0$ m (surface) and $z_2 = 2000$ m. Equation (1) also applies to estimating the heat content of certain water masses within a specific density range, in which $z_1$ and $z_2$ refer to the upper and lower isopycnal surface of that water mass, respectively (Supplementary Figure 1).

Meridional heat advection (MHA, unit: °C/s) was adapted according to Holton and Staley (1973) and was calculated as follows:

$$\text{MHA} = -v\frac{\partial \text{T}}{\partial y} \hspace{5cm} (2)$$

where v is the meridional velocity of an ocean current and the positive velocity indicates a northward current. The $\text{MHA}_{mean}$ represents the regional average of MHA during the whole study period (1979–2019). To compare different extents of

influence of MHA between two periods, MHA changes were calculated: $\text{MHA}_1$ ano, the regional average of the MHA during period 1 (pre-1993) changes relative to the $\text{MHA}_{mean}$; and $\text{MHA}_2$ ano is the regional average of the MHA during period 2 (post-2005) changes relative to the $\text{MHA}_{mean}$. The northward cold-water advection (negative MHA) and southward heat advection (positive MHA) required special attention, as they, simultaneously, signify the influence on OHC and generate changes to water mass properties.

In this study, specifying the position of the SAF was crucial as this boundary manifest interaction between Antarctic and subtropical oceans, determining changes in OHC influenced by meridional heat advection. Previous studies have employed various methods to define the SAF position. The dynamic height method has commonly been used to identify the fronts around the ACC, including the SAF. However, the results obtained from Naveira Garabato (2009) and Kim and Orsi (2014) show a bit of a difference in the position of the front. The cross-track kinetic energy (CKE) was also utilized by Chambers

(2018) to study the frontal system where the SAF shift shows larger amplitudes than the results from Kim and Orsi (2014) and Freeman et al (2016). These differences may result from the complicated influence of eddies and meandering of the ACC. As an important ventilated region, the subpolar Southern Ocean not only contains vigorous air-sea interactions but also involves the complicated influence of ocean currents, compounded further by the strong temperature gradient which lead to the existence of the SAF. Hence, the isotherm has better consistency, similarity, and suitability in describing the SAF

shift and consequent MHA variations. Therefore, in this study, the classical SAF definition was adopted where the SAF is defined as the 4°C isotherm at 200 m (Orsi et al., 1995; Belkin & Gordon, 1996). The SAF shift (unit: ° or km) is described as follows:

$$\text{SAF}_i = \text{SAF}_{period\ i} - \text{SAF}_{mean} \hspace{4cm} (3)$$





$SAF_{mean}$ represents the temporal average position of the SAF. For i = 1, $SAF_1$ is the regional average of the SAF during
period 1 (pre-1993) with changes relative to the $SAF_{mean}$ and i = 2 in $SAF_2$ is the regional average of the SAF during period
2 (post-2005) with changes relative to the $SAF_{mean}$. The regional SAF shift uses two common units: the latitude degree and
the kilometer (km). The regional average of the SAF position, which exhibits a significant shift, is typically described using
the latitude degree unit. This allows for a comprehensive understanding of the overall north-south displacement of the SAF.
When focusing on the specific SAF positions in certain sectors, the kilometer unit is employed to highlight the pronounced
differences at longitudes, providing a more detailed perspective on its spatial features.

The meridional heat advection due to MHA brings about water mass property changes, including isopycnal changes as well
as the displacement of the isopycnals. Therefore, heave-spice decomposition was adopted to properly exhibit the heat
variation, which can reflect the temperature change due to MHA variations and contains two main parts, the spice (isopycnal
component) and heave (the displacement of the isopycnals component) (Zunino et al., 2012; Wang et al., 2021). The Heave-
Spice decomposition is calculated as follows:

$$\left.\frac{dT}{dt}\right|_p = \left.\frac{dT}{dt}\right|_\sigma - \left.\frac{dp}{dt}\right|_\sigma \frac{\partial T}{\partial p} \tag{4}$$

where p is the pressure, T is the conservative temperature, as recent studies (e.g. Groeskamp et al., 2016; Groeskamp et al.,
2019) report that the conservative temperature may be better able to describe heat change even though Zunino et al. (2012)
used the potential temperature. The left term, $\left.\frac{dT}{dt}\right|_p$, denotes the temperature changes under the pressure coordinate. The first
term at the right of the equation (6), $\left.\frac{dT}{dt}\right|_\sigma$, presents the temperature changes caused by spice under the density coordinate.
The second term at the right of the equation (6), $-\left.\frac{dp}{dt}\right|_\sigma \frac{\partial T}{\partial p}$, shows the influence of heave on the temperature changes under
the density coordinate. In this study, we focused on the heat change in heave and spice processes rather than specific water
mass transformation; hence, the terms on the right were not further divided. In addition, as outstanding tilting of isopycnals
exists in the ventilated region and the heat variations in the stratified upper ocean can be revealed in detail by using the
density coordinate, the temperature changes in the heave and spice processes were shown in the density coordinate.

The sea ice variations and the presence of relatively warm subsurface water in the subpolar Southern Ocean have notable
impacts on ocean stratification. Therefore, the stratification index (SI) was appropriate for capturing the influence of cold-
water advection by sea ice and describing surface cooling. SI, as referred to by Margirier et al. (2020), is derived as follows:

$$SI = \int_0^h \frac{g}{\rho_0} \frac{\partial \rho}{\partial z} z \, dz. \tag{5}$$

where g = $9.81\ m \cdot s^{-2}$ is the gravitational acceleration and $\rho_0 = 1025\ kg \cdot m^{-3}$ is the reference density of seawater, z is
the depth, and h is the integral depth. In this study, h = 200 m as surface cooling is most significant within the upper 200 m.
A positive SI confirms the existence of stratification, wherein a larger SI means stronger stratification. This index provides a



quantitative measure of the vertical density gradient and helps to characterize the impact of sea ice variations on surface cooling in regions.

## 3 Results and analysis

### 3.1 Opposite variations of regional heat content

To obtain the pattern and evolutions of subpolar OHC, we first analyzed OHC changes over 2000 m, and spatial/temporal features of regional OHC derived from the observed (MOAA GPV, WOA) and objective analysis (Ishii), as depicted in Figure 1. Overall, subpolar OHC in all basins expressed a steady warming trend, which is consistent with recent studies (e.g. Yang et al., 2020; Wang et al., 2021). This conspicuous increasing trend for subpolar integral OHC can, based on Ishii data, reach 10.8 ZJ/decade (1 ZJ = $10^{21}$ J). Opposite changes were evident in the integral OHC trend of the subpolar Southern Ocean. Opposing trends were observed in the southwest Pacific (52–66°S, 130–170°W, hereafter referred to as the 'Pac' sector) and the South Atlantic and Indian (50°W–62°S, 40°W–120°E, hereafter referred to as the 'Atl–Ind' sector). The Atl–Ind sector had the most warming, reaching 2.2 ZJ/decade, and accounted for around 20.4% of subpolar warming, implying that the Atl–Ind sector and nearby regions contributed more than 1/5 of warming to the circumpolar OHC. Although unexpected cooling was limited in the Pac sector, it still attained −0.1 ZJ/decade. All the datasets demonstrated that the opposing changes have been remarkable over the past few decades. These subpolar OHC changes described by Ishii are supported by the observed data. The correlation coefficient of integral OHC over 2000 m between Ishii and MOAA GPV was 0.92 in the Atl–Ind sector and 0.63 in the Pac sector during the same period. Supplementary Figures 2, 3 also presented similar regional-mean temperature and salinity profiles for Ishii data with WOD 18. The opposing changes were also steadily sustained during the evolutions of the subpolar OHC, showing discernible interannual variations imposed on the long-trend trend. This change became more significant during the Argo period. Thus, the study period from 1979 to 2019 for comparing OHC changes (Figure 1d) was divided reasonably into two periods: pre-1993 (period 1) and post-2005 (period 2). We also checked the wavelet analysis and power spectrum analysis (not shown) in the OHC time series and found that 16 months was ideal for removing seasonal interference. The regional OHC changes in the original and filtered time series both showed significant differences between periods 1 and 2. In addition, we also checked OHC variations of the two primary water masses (not shown) in the subpolar ocean, AAIW (27.0 < σ < 27.5 kg/m$^3$, Portela et al., 2020) and UCDW (27.5 < σ < 27.8 kg/m$^3$, Silvester et al., 2014). OHC changes in AAIW and UCDW also exhibited significant warming in the Atl–Ind sector (AAIW: 0.4 ZJ/decade; UCDW: 2.5 ZJ/decade) and similar cooling in the Pacific sector (AAIW: −0.1 ZJ/decade; UCDW: −0.2 ZJ/decade), see Supplementary Figure 1.

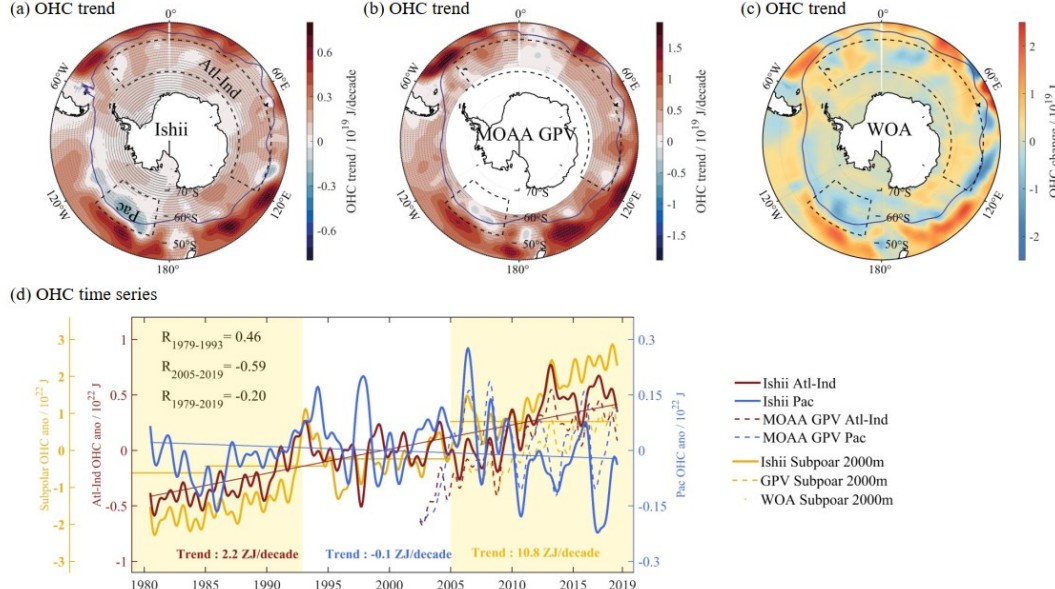

**Figure 1: The OHC temporal and spatial variation. a) The vertical integral OHC trends (unit: $10^{19}$ J/decade) above 2000 m derived from Ishii during 1979–2019. The dashed black frames with tags 'Atl–Ind' and 'Pac' represent the regions with significant OHC trends in the Atlantic–Indian (50–62°S, 40°W–120°E) and Pacific sectors (52–66°S, 130–170°W), respectively. The stippling denotes the OHC trend exceeding the 95% confidence level from the two-tailed $t$ test. The purple line represents the climatological SAF. b) same as a) but from MOAA GPV data from 2001–2019. c) The OHC changes (unit: $10^{19}$ J) derived from WOA between the former (pre-1993) and latter (post-2005) periods. The OHC change is calculated as OHC $_{pre-1993}$ minus OHC $_{post-2005}$. d) The time series of OHC anomaly (unit: $10^{22}$ J) in the Atl–Ind and Pac sectors, and circumpolar > 2000 m. The dashed, thick solid, and thin solid lines denote the OHC anomalies from MOAA GPV, Ishii, and WOA, respectively. The straight lines show the linear trend corresponding to each time series with the same color. The yellow lines denote the circumpolar OHC anomalies of the subpolar Southern Ocean (45–65°S, 0–360°).**

The opposing changes, as shown in Figure 2, also exhibited pronounced inverse changes between the Atl–Ind and Pac sectors from pre-1993 to post-2005. In view of the severe tilt of isopycnals and strong ventilation in the subpolar ocean, the density coordinate was more suitable than the pressure coordinate for showing vertical changes. In the Atl–Ind sector, from period 1 to period 2, the most significant changes occurred around 26.8 kg/m³ with 0.4°C warming and 0.02 psu salinization. On the contrary, there was mild cooling (maximum temperature change: 0.04°C) and weak changes in freshening (maximum salinity change: −0.003 psu) between 26.9 kg/m³ and 27.1 kg/m³ (Figure 2a), reflective of the influence of surface in the ventilated regions. Overall, this cooling and freshening covered the subpolar region and was evident in a thin and shallow layer of < 100 m. Warming and salinization were notable again below the mixed layer within 26.9 and 27.1 kg/m³ and the strongest temperature (salinity) change was 0.14°C (0.01 psu). The most significant warming and salinization trend peaked at 0.14°C/decade and 0.01 psu/decade (Figure 2c), respectively. These significant trends were consistent with Yang et al (2020) and both occurred at 200 m where the temperature and salinity changes were closely related to the heat advection. However, this cooling appeared to have been limited at the surface due to robust stratification (detailed in Section 3.2).




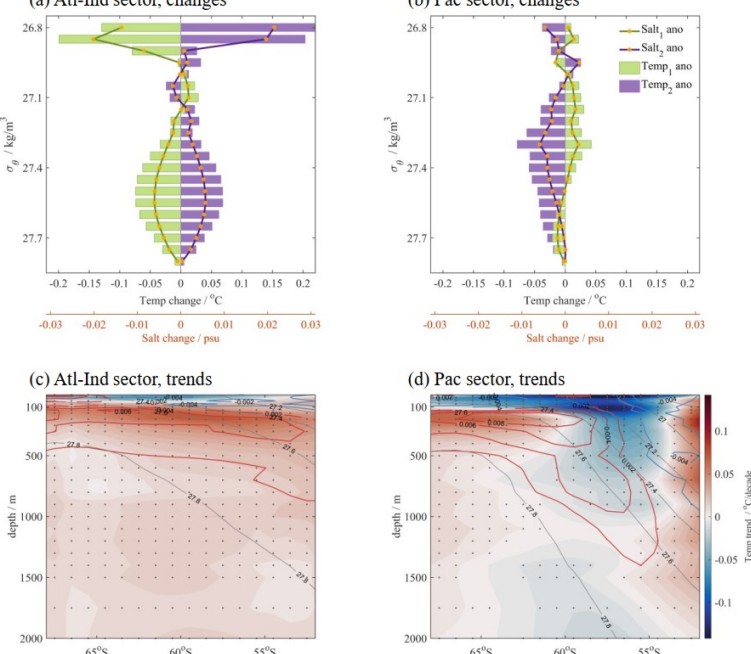

**Figure 2: The vertical structure of the regional mean temperature and salinity change based on Ishii. a) The vertical temperature and salinity anomalies in the Atl–Ind sector in the potential density coordinate. The green (purple) bars represent the temperature difference between the mean temperature of period 1 (period 2) and the climatological temperature of the whole study period. This temperature difference is calculated as $T_{period\ 1}$- $T_{climatology}$ and $T_{period\ 2}$- $T_{climatology}$. The green (purple) lines with yellow circles represent salinity difference calculated as temperature difference. The temperature change is on the black x-axis (unit: °C) and the salinity change is on the orange x-axis (unit: psu). The interval of potential density on the y-axis is 0.05 kg/m³. b) same as a) but in Pac sector. c) Temperature trend (unit: °C/decade) and salinity trend (unit: psu/decade) in the Atl–Ind section. The shadow and contours donate the temperature trend and salinity trend, respectively. The dotted shows the OHC trend exceeding 95% significant level. d) same as c) but in the Pac sector.**

Comparatively, temperature and salinity in the Pac sector show inverse changes in general than those in the Atl–Ind sector. From period 1 to period 2, the strongest cooling and freshening occurred at 27.3 kg/m³ with −0.12°C and −0.01 psu. There was also significant surface cooling, consistent with Kang et al (2023). Contrary to cooling evident in the Atl–Ind sector, this surface cooling trend in the Pacific peaked at −0.16°C/decade and covered water mass within 27.0 and 27.6 kg/m³, where most freshening trend occurred (−0.005 psu/decade). This northward cold and fresh water was capable of penetrating deeper layers, which manifests in stronger isopycnals tilting (Figure 2d), indicating that the isopycnal processes as well as the displacement of the isopycnals can facilitate more intense cold-water intrusion to the interior of the Pacific. These temperature changes also indicate the influence of the heave and spice process due to changes in MHA. Notably, there was also significant subsurface warming (0.18°C/decade) and salinization (0.01 psu/decade) in the Pac sector (60–66°S, 100 m < depth < 500 m), where the salinization trend appeared to spread from the higher latitudes (north of 65°S) to the subpolar regions (55–60°S), suggesting that the northward advections also generated northward warm subsurface water. Consequently, the vertical opposing changes, which were more discernible in the density coordinate than the pressure coordinate, are also more significant and complex. These significant vertical opposing changes were characterized by a clear warming–cooling–



warming change in the Atl–Ind sector; however, a cooling-warming-cooling trend can be seen in the Pac sector. These spatial/temporal opposing changes between two sectors demonstrate that there were different influences of meridional heat advection, indicating possible opposing changes in the Antarctic-Subtropical boundary, which is shown by changes in the

SAF shift (Figure 3).

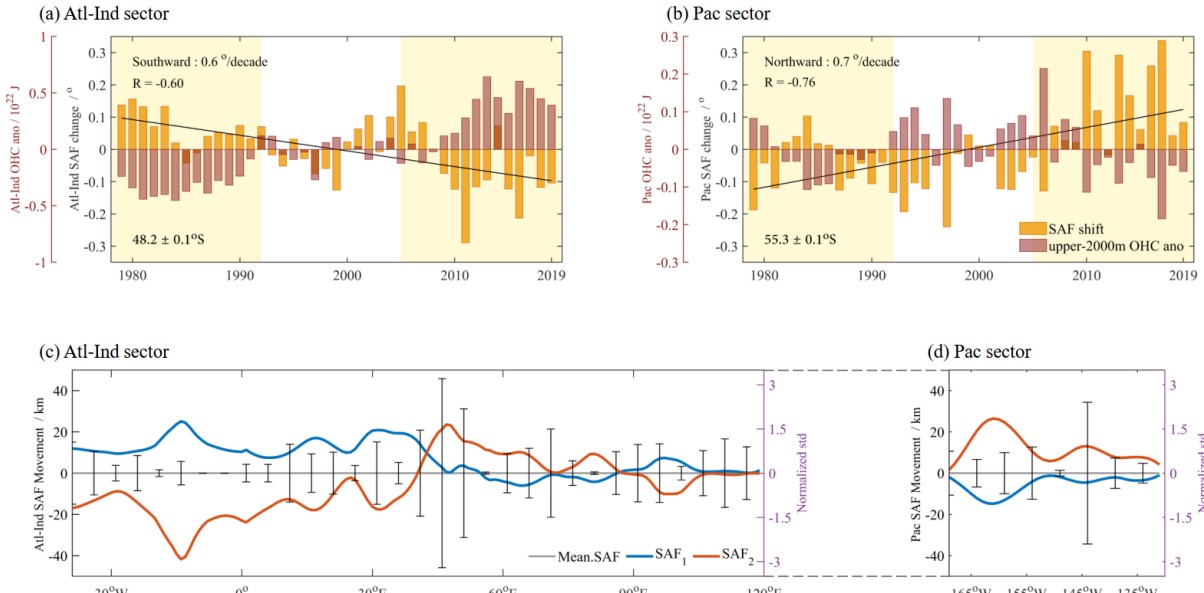

**Figure 3: SAF variations. a) The annual time series of regional mean SAF shift shown by orange bars using black y-axis in Atl–Ind sector with the regional mean position and standard deviation in the top left. The black line is the linear trend (unit: °/decade) of the SAF. The brown bars denote the upper-2000 m OHC anomalies using the left-blown y-axis. b) same as a) but in Pac sector. c)**
**The SAF shift (unit: km). Note that the SAF shift uses the movement distance (kilometres) rather than latitude and the shift exhibits clear in region. The movement of SAF in period 1 and period 2 relative to the spatial and temporal average SAF (Mean SAF, the black zero line) are shown by blue and red lines, respectively. The error bars show the normalized standard deviation in each longitude and use the purple y-axis on the right. The interval of SAF longitude is 5°. The dashed lines between c) and d) denote the subpolar region not shown.**

The southward SAF indicated influence by more heat advection from Subtropical into Antarctic, and the northward SAF means more cold-water advection from Antarctic into Subtropical. The apparent warming amplitudes in the Atl–Ind sector that occurred during 2012–2013 (Figure 1) coincided with the southmost shift of the SAF (Freeman et al., 2016), suggesting a possible correlation between the poleward MHA and the occurrence of increasing subpolar OHC. These apparent warming amplitudes also occurred during 2012–2013 as the SAF had the southmost shift. In the Pacific sector, the more pronounced

decrease in OHC during 2015–2016 occurred at the maximum Antarctic sea ice melt (Wang et al., 2019), indicating that enough melting sea ice can supply sufficient cold water. Hence, determining the role that the MHA plays in the formation of the prominent and stable subpolar OHC dipole, particularly in the context of opposing changes, requires an understanding of the evolution and regional characteristics of the SAF shift. Here, the annual SAF time series is used to avoid interference from the potentially more distinct (i.e., larger amplitude) seasonal influence. The circumpolar SAF also exhibited distinct

inverse changes with steady trends in the regions, as well as a southward shift in the Atl–Ind sector and a northward shift in





the Pac sector. The mean SAF in the Atl–Ind sector was around 48.2 ± 0.1°S and showed the most significant southward trend (−0.6°/decade) (Figure 3a). The SAF shifted significantly southward by 0.1°, between periods 1–2, which was consistent with several studies (e.g., Downes et al., 2011; Freeman et al., 2016). This poleward shift of the SAF coincided with increasing OHC where both variables had high correlation coefficients of −0.60 (OHC above 2000 m and the SAF), −.67 (AAIW OHC and the SAF) and −0.54 (UCDW OHC and the SAF) (Table 1), implicit of the influence by poleward heat advection on OHC trends. These negative correlation coefficients mean that the southward SAF (decreasing latitude) always occurs with the increase of the OHC. The correlation coefficients between OHC and MHA are satisfactory and reached 0.40 with the OHC above 2000 m and 0.44 with the heat content of the UCDW. Between periods 1–2, the SAF showed the most significant southward shift (around 66.8 km) at 14°W. Though the SAF still shifted southward at most longitudes, there was also a slight northward change between 43 and 87°E where the amplitude of the SAF was the greatest and reached 131.4 km (Figure 3c). Similarly, OHC in each sector, similar to the SAF shift, showed clear changes between the two periods.

**Table 1. The correlation coefficients in the Atl–Ind sector and Pac sector. The upper panels are the correlation coefficients between OHC and SAF, MHA. The lower panel is the correlation coefficient between SAF and MHA. The star* in the top right corner of the coefficient means not significant (Alpha = 0.05). The time series used are annual times series to avoid interfering with the seasonal influence with possible larger amplitude.**

| Correlation with OHC | | Atl-Ind sector | Pac sector |
| --- | --- | --- | --- |
| Upper-2000 m | | -0.60 | -0.76 |
| AAIW | SAF | -0.67 | -0.83 |
| UCDW | | -0.54 | -0.52 |
| Upper-2000 m | | 0.40 | 0.16* |
| AAIW | MHA | 0.14* | 0.30 |
| UCDW | | 0.44 | 0.10* |
| Correlation between SAF and MHA | | Atl-Ind sector | Pac sector |
| SAF | MHA | -0.32 | -0.45 |

The Pacific sector had the southernmost part of the SAF, the part existed at 55.3 ± 0.1°S of the circumpolar SAF. The SAF showed an inverse trend compared to the trend in the Atl–Ind sector. This northward trend was a shift of 0.7°/decade (Figure 3b) with a 0.2° shift to the equator from period 1 to period 2. This northward trend was substantial throughout the whole study period, displaying a consistent northward shift at all longitudes where the northmost shift was 41.3 km at 161°W. The greatest amplitude was 85.6 km and occurred at 144°W (Figure 3d). This northward SAF in the Pac sector also showed a negative correlation with the decreasing OHC, where the correlation coefficients were −0.76 with the OHC above 2000 m, −0.83 with the heat content in the AAIW, and −0.52 with the heat content of the UCDW (Table 1). These negative correlation coefficients in the Pac sector denoted that a northward SAF (increasing latitude) occurred with decreasing OHC. These satisfactory correlations between OHC changes and SAF shift mean that there was prominent meridional heat and/or cold advection.

### 3.2 Regional MHA changes driven by wind anomalies

Before explaining the influence of heat advection on OHC, the recent changes in meridional heat advection need to be clearly described. Since the SAF in the Atl–Ind and the Pac sectors displayed an inverse change and the variations of the net



regional mean MHA were weak as well, the regional mean southward MHA and northward MHA were more appropriate for
describing each influence of the MHA in the Atl–Ind sector and Pac sector, respectively. The spatial and temporal MHA
averages emphasized in two sectors are shown in Figure 4a. Similar to OHC and the SAF, MHA in the two sectors changed
significantly from period 2 to period 1 (Figure 4b). The southward MHA represents a mild increasing trend ($0.1 \times 10^{-7}$ °C/m/decade) and its anomalies express a stronger increase before filtering in period 2 compared to those in period 1.
In contrast, the northward MHA (cold-water advection) also enhanced significantly, showing a strong trend ($-0.2 \times 10^{-7}$ °C/m/decade). These significant trends indicate that the strong meridional heat/cold-water advection occurring with
southward/northward SAF shift and subsequently, an increase/decrease in regional OHC. The strongest MHA anomalies
occurred between 26.8 and 27.4 kg/m$^3$ (Figure 4c, d) which covered the significant ventilated isopycnals in the subpolar
region (Figure 2). The southward MHA in the Atl–Ind sector increased significantly to $2.2 \times 10^{-7}$ °C/m/decade at
approximately 26.8 kg/m$^3$ between periods 1–2 and steadily became more enhanced in the vertical direction, yet this
intensification weakened with increasing depth (Figure 4c). Similarly, the cold advection in the Pac sector also enhanced to
$2.5 \times 10^{-7}$ °C/m/decade in a northward direction between periods 1–2 at 26.8 kg/m$^3$ (Figure 4d). Although apparent
changes in heat and cold-water advections seemed to be restricted in the near surface layer, these advection anomalies can
still explain the impact of MHA on OHC, as the influence could penetrate deeper by ventilation and isopycnal processes or
the displacement of the isopycnals. These processes of heave and spice are explained in section 3.3 in detail.

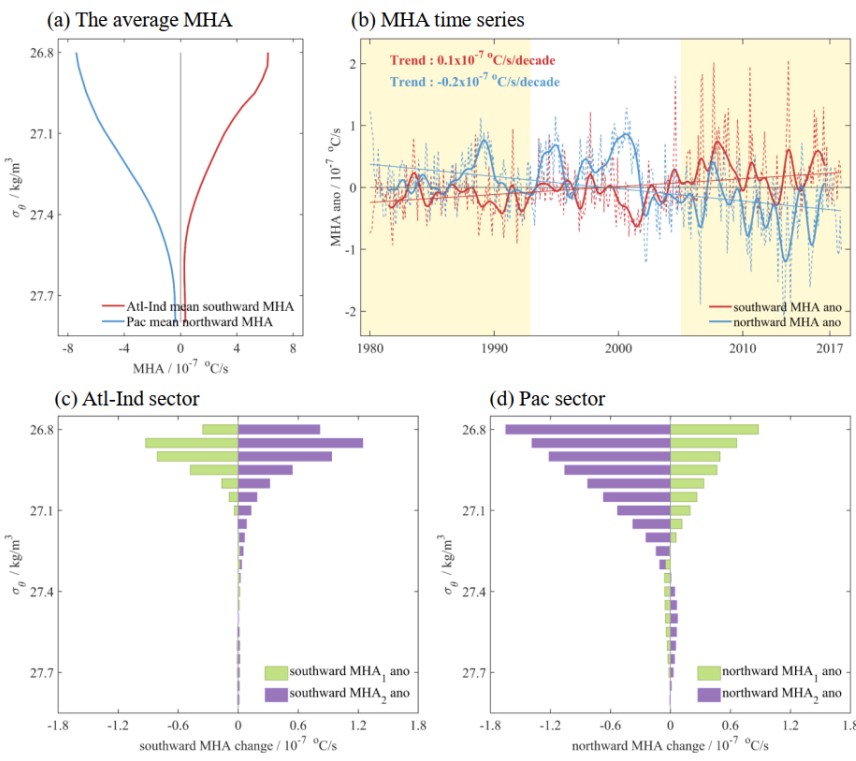



**Figure 4: MHA variations. a) The temporal and spatial mean MHA profiles. b) The MHA time series in each sector. The red line and blue line denote the temporal and spatial southward MHA in the Atl–Ind sector and northward MHA in the Pac sector, respectively. The straight lines are the corresponding linear trends of the time series with the same color. And trends are shown in the top left. The monthly MHA anomalies shown by solid lines are filtered by 16 months to better remove seasonal interference and the dashed lines are initial anomalies before filtering. c) The southward MHA change in the Atl–Ind sector. The green (purple) bars represent the MHA difference between the mean MHA of period 1 (period 2) and the mean MHA of the whole study period. d) same to c) but in Pac sector.**

The correlations between SAF, MHA, and OHC further explain the conspicuous meridional advection resulting from the significant SAF shift, which subsequently, affected OHC variations (Figure 5). Overall, the meridional heat and cold-water advections are significant during shifting SAF which were shown in resultant correlation coefficients with the SAF; −0.32 in the Atl–Ind sector and -0.45 in the Pac sector (Table 1). Notably, the positive (negative) anomalies mean there was an enhancement of the southward (northward) MHA in the Atl–Ind (Pac) sector (Figure 5a, b). The southward SAF in the Atl–Ind sector ensured enhanced southward MHA (Figure 5a). The most significant MHA changes occurred at 26.8–27.1 kg/m³ and southward MHA increased conspicuously during the southward SAF compared to the northward SAF. Similar warming influence from the poleward shift of fronts in the water mass was also reported by previous studies (e.g. Meijers et al., 2011). However, in this study, we explored further that, the SAF shift represents the interaction between Antarctic and Subtropical, and consequently, the MHA had a more important influence in modulating regional the heat redistribution and modulated the upper 2000 m OHC changes.

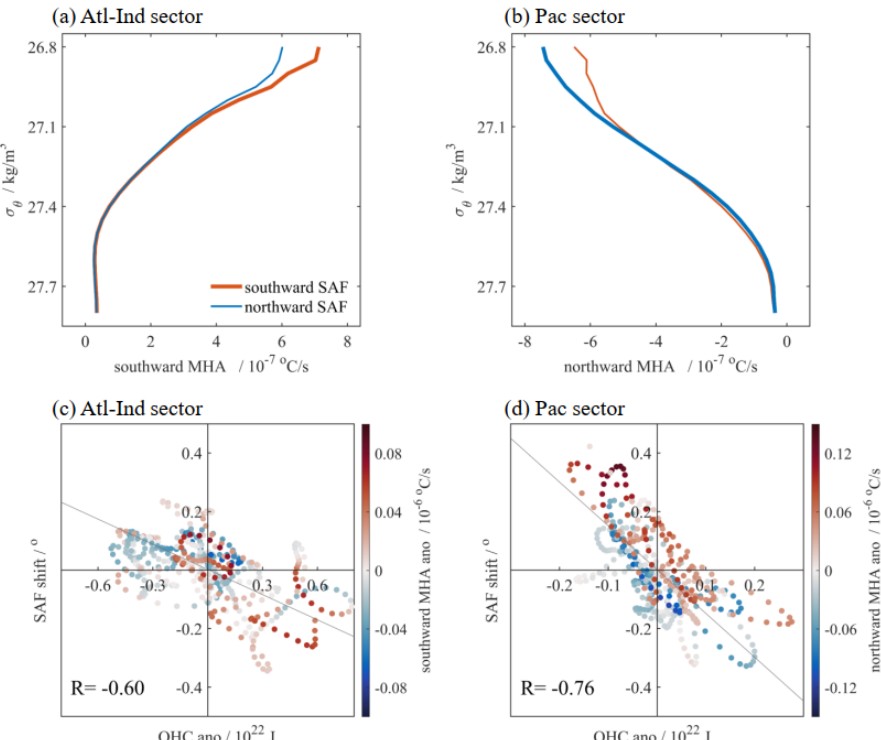

**Figure 5: MHA change associated with SAF shift and OHC anomalies. a) The mean MHA compositions in the Atl–Ind sector. The red and blue lines denote the MHA profiles during the southward and northward SAF, respectively. b) same as a) but in the Pac sector. The dominant advections in each sector are emphasized by the thick line. c) The southward MHA anomalies associated**





**with the SAF shifting and OHC anomalies in the Atl–Ind sector. The colour scatters represent the southward MHA anomalies > 2000 m and the positive value denotes the increasing southward MHA. The correlation coefficient between SAF and OHC is at the left bottom. d) the same as c) but for the northward MHA anomalies in the Pac sector. The gray lines in c) and d) are the linear fitting by the scatter.**

The increases in OHC in the Atl–Ind sector during the increase of the southward MHA and OHC showed a good correlation with the poleward MHA, where the coefficients were 0.40 (MHA and upper 2000 m OHC) and 0.44 (MHA and UCDW heat content) (Table 1). The correlation coefficient between MHA and the heat content of AAIW was not significant (R = 0.14) since MHA increased while the light AAIW cooled slightly, which was especially significantly from 27.0–27.2 kg/m$^3$ (Figure 2a) in the Atl–Ind sector, which could be explained by spice processes (section 3.3). An increase in OHC still occurred with stronger southward MHA during the poleward SAF during most periods in the Atl–Ind sector. Comparatively, in the Pac sector, northward SAF dominated and caused enhanced cold-water advection at the surface (26.8–27.2 kg/m$^3$), while a mild increase in the MHA occurred at the subsurface (27.3–27.8 kg/m$^3$, Figure 5b). This subsurface warming resulted from the northward subsurface warm water (Figure 2d) during northward SAF. Unfortunately, correlations between Pacific MHA and heat content (upper 2000 m: 0.16*; AAIW: 0.30; UCDW: 0.10*) failed to be satisfactory (Table 1), implying that Pacific cooling was complicated and may work at longer time scale. Nevertheless, stronger cooling in the Pac sector still mostly occurred with northward cold-water advection as the SAF shifted towards the equator (Figure 4d).

Fronts movement reflected the heat exchange between Antarctic and subtropical in the upper ocean, which was influence of meridional advections modulated by the wind field over the Southern Ocean. With the development of enhanced atmospheric circulation over the Southern Ocean, wind field changed heat exchange between high and mid latitudes in the atmosphere as well as in the ocean and consequently, changes in regional OHC (Figure 6). In the Atl–Ind sector, the north positive wind curl strengthened from pre-1993 to post-2005 (Figure 6a), corresponding to the subtropical gyre below. The maximum of positive wind curl anomalies got one and a half times larger in post-2005 than pre-1993, and the longest duration of positive wind curl anomalies changed from 12 months in pre-1993 to 14 months in post-2005, and the longest duration was 24 months occurring during 1993-2005 (Figure 6b), implying the southward MHA from subtropical got enhanced and also occurred frequently after 1993. The evolution of positive wind curl anomalies was also consistent with the running correlation coefficients (Figure 6c). Positive RCC behaved well after 2005 that means the increasing MHA resulting from the enhanced wind over the subtropical gyre. The westerly wind, south branch of the anticlockwise wind field over subtropical gyre, got enhanced and shifted poleward, carrying heat to the subpolar by the means of increasing MHA. Then, the regional OHC increased.





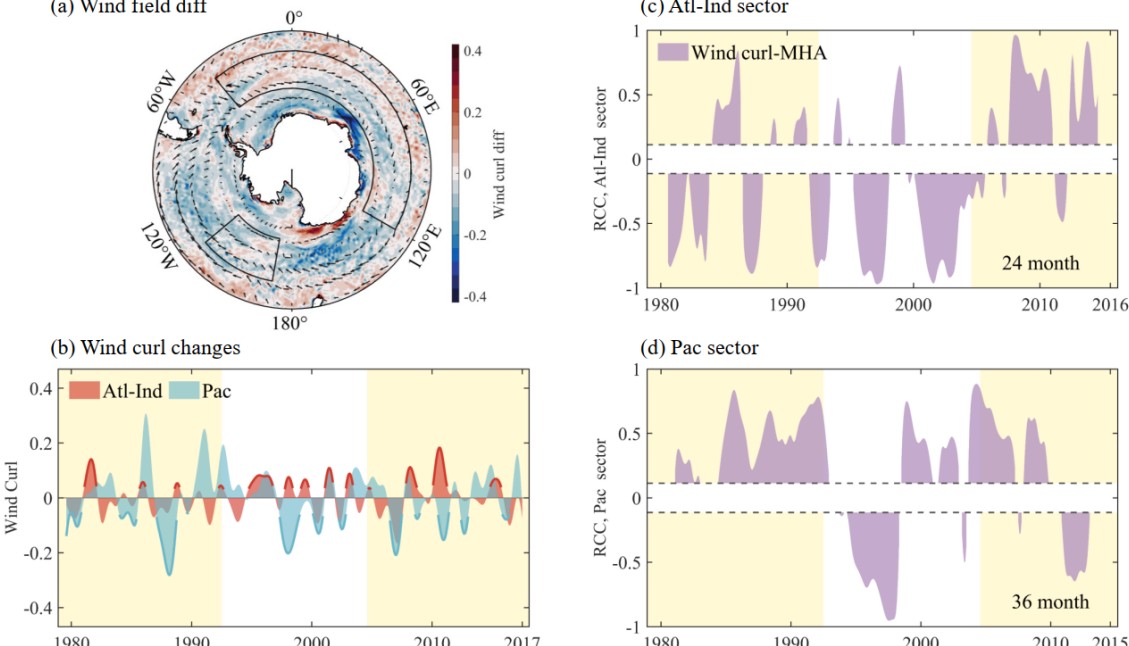

**Figure 6: Wind field changes and running correlation coefficients (RCC) between Wind curl and MHA. a) Time series of wind curl changes. The red and blue shadows denote anomalies of the Atl–Ind sector and Pac sector, respectively. The arrows denote changes of wind field (2005–2019 minus 1979–1993). b) c) Time series of RCC in the Atl–Ind sector. d) same to c) but in the Pac sector. The blank shows the RCC failed to exceed the 95% confidence level from the two-tailed t test.**

But in the Pac sector, advections were associated with negative wind curl anomalies, which was dominated in this sector (Figure 6b). The negative wind curl anomalies also occupied frequently for long periods of time and the occurrence was increased by 50% after 1992 than before. What's more, the positive anomalies were reduced significantly from pre-1993 to post-2005. And the amplitude also decreased by 85% (Figure 6b), implying the enhance of the negative wind curl. Different from the wind field in the Atl–Ind sector, there was dominated atmospheric circulation, Amundsen Sea Low (ASL) over the south Pacific. The persistent positive RCC demonstrated strong influence by ASL (Figure 6d). The northward of ASL existed in the Pac sector, and the negative anomalies of wind curl were corresponding to the enchanted low pressure in recent years which reinforced northward meridional exchange from the high to the mid latitudes in the atmosphere as well as in the upper ocean. Hence, the cold-water advection went northward accompanied by equatorward SAF. Additionally, there were also negative RCC at 1994–1998 and 2011–2013 and were affected by the decreasing ASL (positive wind curl anomalies in Figure 6b). Hence, wind filed anomalies played an important role in modulating the meridional advection. Consequently, changes in regional OHC.

### 3.3 Water mass property changes

Regional OHC variations signified the property changes in water mass and the heat/cold-water advection resulted in water mass property changes. Changes in water mass properties are shown by the arrows, consisting of the salinity trend (x-vector)





and the temperature trend (y-vector) corresponding to specific patches with temperature and salinity limitations (Figure 7). The arrows describe the property variations due to advections during SAF shift rather than the specific water mass changes between different parts; in that, the water mass property changes in this study were different from the water mass

transformation in other studies (e.g. Groeskamp et al., 2016; Zika et al., 2021). The patches with colors and without arrows in Figure 7 indicate that the salinity trend failed to pass the 95% confidence test. These property changes showed significant differences due to the different MHA influence and also exhibited a distinction between periods 1–2. In the Atl–Ind sector, only the ventilated fresher water (about 27.2 kg/m³, S < 34.1) near the subpolar surface presented isolated cooling in period 1 (Figure 7a). This cooling part had the most significant isopycnal change. This limited cooling completely disappeared and

subsequently, intense warming covered the whole upper 2000 m during period 2 (Figure 7c) when southward heat advection was strong during southward SAF. Most warming of significance occurred mainly in two regions: one was towards the north of the subpolar, having a potential density of around 26.9 kg/m³, which was similar to the study by Yang et al., (2020). Another warmed area was in the deep ocean near 27.8 kg/m³, consistent with increasing OHC trend (Figure 1). In the view of good correlations between SAF movement and MHA changes, we used the compositions during different SAF shift to

reflect strong/weak influence by MHA on water mass property changes (Figure 7e, f). During the southward SAF, coverage of warming occupied almost all patches (Figure 7e), indicating that the warming intruded near the ventilated regions resulting from increasing heat advection during the poleward SAF.

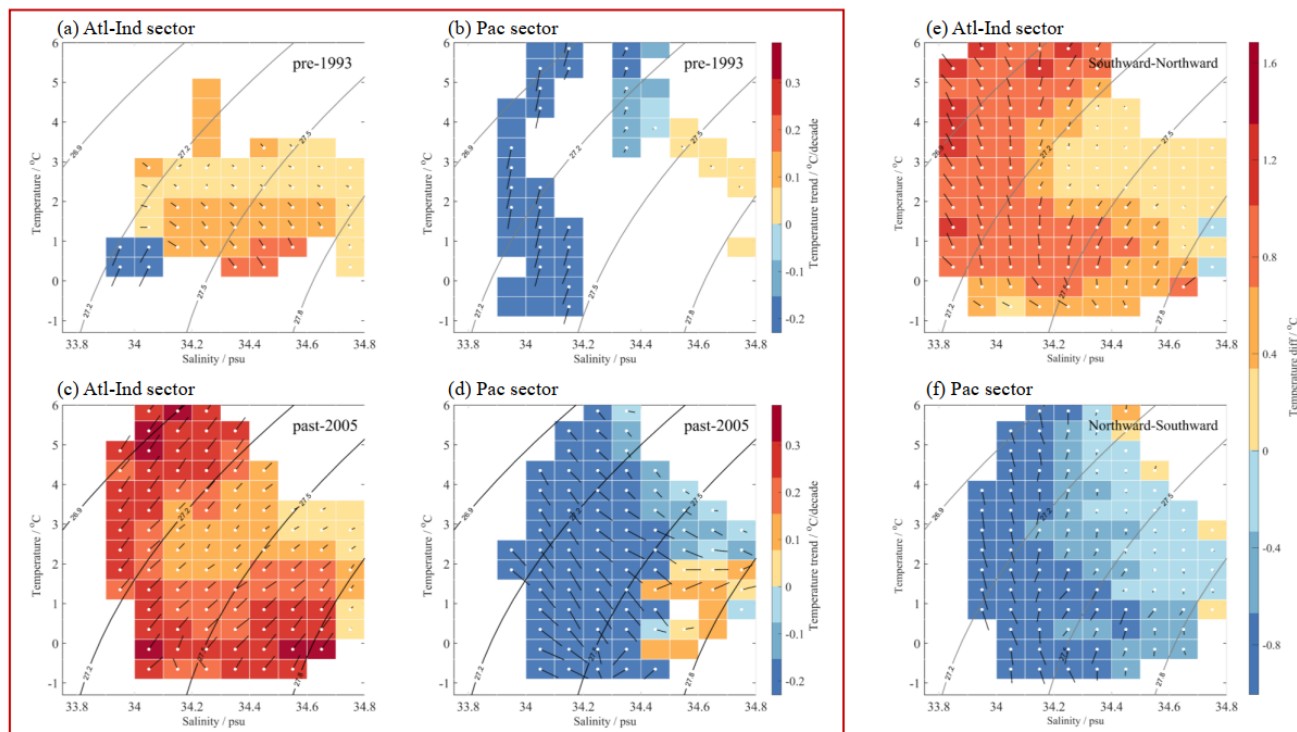

**Figure 7:The water mass property change. a) The Temperature and salinity trends in the Atl–Ind sector during period 1. The**
420 **patches in the upper (lower) panel of the left red box show the temperature trends at a 95% level in period 1 (period 2). The lost**



**patches are the regions in which their trends fail to be significant or they are ventilated sometimes and the ventilated duration is larger than 1/3 of the whole study period, thus these trends are blanked. The arrows with white dots at the start represent the direction of water mass property change. The x-vector and y-vector of the arrows are temperature and salinity trends during each period, respectively. b) same as a) but in Pac sector. c) same to a), d) same to b) but in period 2. e) the temperature difference from the northward SAF to the southward SAF in the Atl–Ind sector is shown by the patches. f) same to e), but those difference is from the southward SAF to the northward SAF. Note that these water mass property changes denote temperature and salinity changes or trends in this paper rather than the water mass transformation (e.g. Groeskamp et al., 2016; Zika et al., 2021).**

In comparison to the warming in the Atl–Ind sector, remarkable cooling occupied almost all water masses in the Pac sector post-2005, as well as the northward SAF period (Figure 7b, d). It is important to note that there are many blanks in Figure 7b, d which denote the regions where patches in temperature and salinity trends failed to be significant or were, at times, ventilated over a duration larger than 1/3 of the whole study duration, thus these trends were not included. Nevertheless, widespread cooling, generally, displayed an intensified trend above 27.5 kg/m$^3$ and there was also mild warming between 27.5 and 27.8 kg/m$^3$. This warming below 27.5 kg/m$^3$ in period 2 was different from that shown in Figure 7f. Thus, this warming seems to be influenced by the upwelling of warm UCDW, mentioned in numerous previous studies (e.g. Haumann et al., 2020; Purich and England, 2021; Herraiz-Borreguero and Naveira Garabato, 2022), that was due to enhanced westerly wind post-2005 rather than subsurface warm water (Figure 7f). This upwelling of UCDW indicates that there was important displacement of the isopycnal process, heave, in the deep and conspicuous isopycnal process, spice, due to strong stratification from the surface to the deep layers (shown in Figure 9).

To further explain the influence of cold-water advection in the Pac sector, it was necessary to assess sea ice variations, which enhanced the Pacific surface cooling. Surface cooling existed in both the Atl–Ind sector and the Pac sector but was more significant in the latter region and penetrated deeper layers, causing more severe cooling in deep ocean (Figure 2). The stratification index (SI) was employed to express the strength of stratification, where the larger SI reflects more surface cold-water and stronger cold-water advection by the SIC. Cooling was significant within the upper 200 m (Figure 8), thus, in this study, h = 200 m. The positive SI confirms the existence of stratification and the larger SI means stronger stratification. Notably, larger SI means a weaker diapycnal process but do not mean also a weak isopycnal process. In other words, isopycnal processes could be still stronger if there were sufficient cold-water influx by the accumulation of sea ice melt in the surface, indicating stronger spice processes. In the Pac sector, when the SAF shifted toward the equator, surface cooling was enhanced by the increasing SIC (the maximum increase reached 68.4%). The strong negative correlation between stratification and surface temperature changes had a satisfactory coefficient at −0.63 (Figure 8d), and the correlation coefficient between SIC and surface temperature was also significant (R = 0.86), signifying that the weaker the stratification (smaller SI), the more there may be sea ice, and the stronger the cooling (Figures 8b, d). More explicitly, during the northward SAF periods, the increasing SIC in the Pac sector was more capable of supplying cold-water advection via melting as well as transport. Following this, the abundant cold-water accumulated at the surface led to the sink of the isopycnals. At the same time, the stronger tilting of isopycnals in the Pac sector (Figure 8b) also facilitated the deeper cold-water penetration, indicating that cooling invasion by spice processes may play an important role in facilitating surface cooling via isopycnal process.



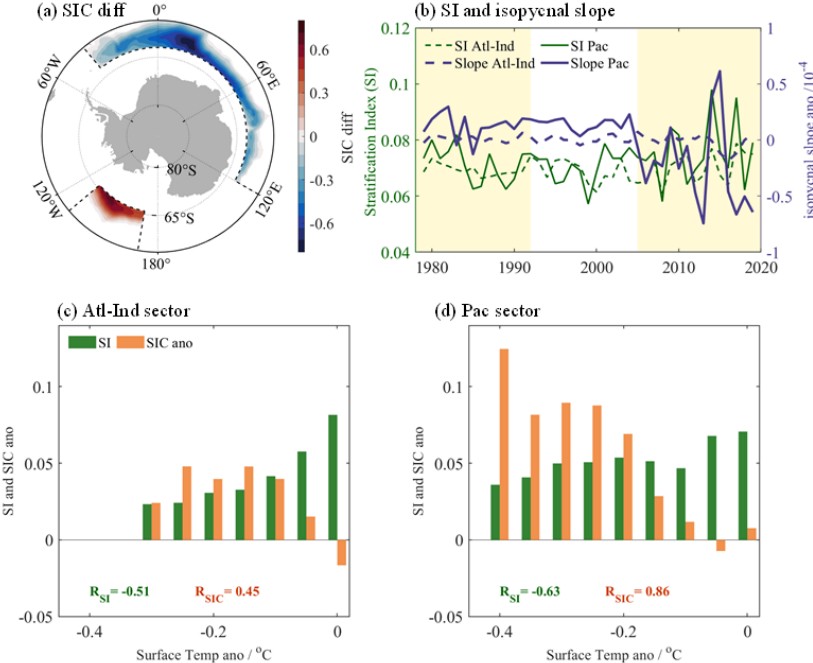

**Figure 8: Surface cooling associated with SIC variations. a) The SIC anomalies in the Atl–Ind sector are calculated SIC southward SAF-SIC northward SAF. While the SIC anomalies in the Pac sector are calculated SIC northward SAF-SIC southward SAF. b) the time series of the stratification index (SI) and isopycnal slope anomalies. The dashed line and solid line denote the Atl–Ind sector and the Pac sector, respectively. The green and purple lines present the SI and slope, respectively. c) The SI and SIC changes with the surface temperature. The green and orange bars present the SI and the SIC anomalies, respectively. The SI is shown clearly by five times magnification for using the same y-axis with the SIC anomalies. The correlation coefficients with the same color show the corresponding correlation with the surface temperature changes.**

Comparatively, in the Atl–Ind sector, decreases in sea ice occurred during the southward SAF and the coefficients between the stratification and SIC with surface temperature were −0.51 and 0.45 (Figure 8c), respectively; both of which were relatively weak compared to those in Pac sector. In that, the heat induced by poleward advection led to decreasing SIC with a maximum decrease of 75.7% (Figure 8a). At the same time, there was strong stratification and relatively weak isopycnal tilting compared to the Pac sector. Therefore, surface cooling in the Atl–Ind sector was restricted to a thin, shallow layer, implying that there was weak intrusion of surface cold water (Figure 8b) and that cooling-inducing spice dominated near the ventilated surface (Figure 9b) while heave may have played a more important role in the deep as the upwelling of the warm UCDW driven by enhanced westerly wind (Figure 6a).

Changes in temperature and salinity of water mass were closely related to isopycnal process (Zunino et al., 2012). As mentioned above, these significant temperature and salinity changes mean that the influence of heat advection may result in strong isopycnal processes. Hence, heat variations modulated by advection were described from two perspectives: the temperature changes on isopycnals were presented by the spicing process, and temperature changes due to the displacement of isopycnals were displayed by the heaving process (Figure 9). The temperature trends in heave and spice components were calculated by equation (4). The blanks in Figure 9 are the regions with intermittent/periodic ventilation. In the Atl–Ind sector,




significant warming occurred in most isopycnal layers, especially 27.4–27.8 kg/m³, where the thick AAIW and warm
UCDW were located, manifesting the influence by the upwelling of the warm UCDW via heave process. Notably, the
influence of heaving on surface cooling in Figure 9a appears to be different from that in Supplementary Figure 5b since
ocean takes in surface heat flux due to deepening of isopycnals induced by wind curl during the whole study period. Spice
process behaves cooling during southward SAF (Figure 9b) and warming during 1979–2019 (Supplementary Figure 5c).
That is also understandable, as the poleward heat advection induced sea ice melting and caused subsequent surface cooling.
At the same time, the subsequent poleward heat advection also caused apparent isopycnal warming/spice process in deep
layers (Figure 9b). This cooling also partly resulted from the surface heat loss near the ventilated region. Overall, in the Atl–
Ind sector, the compounding of heave and spice processes gave rise to the intense warming.

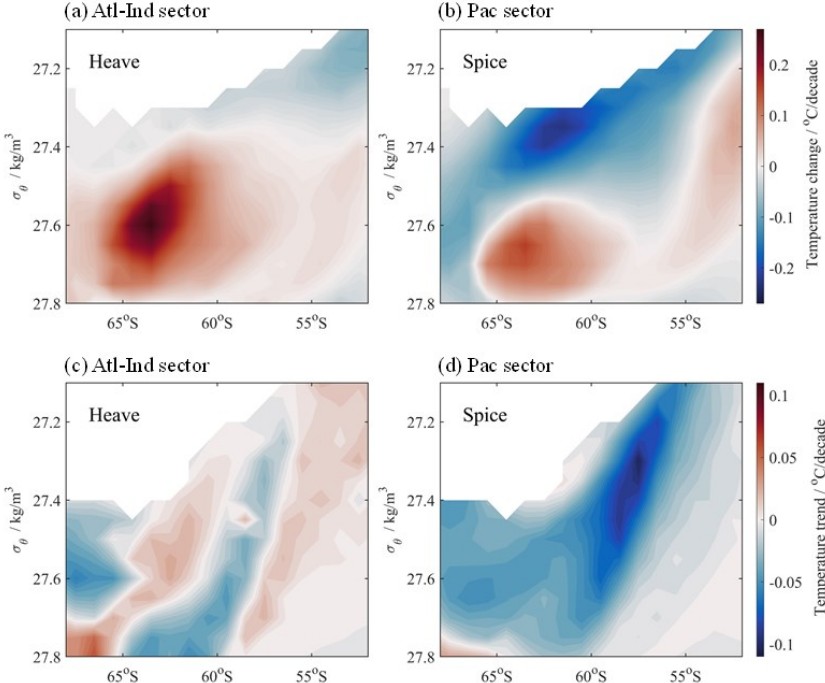

**Figure 9: Temperature change by the heave and spice. a) The temperature trend (unit: °C/decade) by the heave process during the**
**southward SAF in the Atl–Ind sector. b) same as a) but of the spice process. c) same to a), d) same to b) but during the northward**
**SAF in the Pac sector.**

But cooling in Pacific was mainly generated from spice processes as the heat advection went towards the equator (Figure 9d).
There was a strong cooling trend reaching −0.12°C/decade which occurred along all isopycnals between 57 and 62°S with a
few surrounding weak warming centers (Supplementary Figure 5d). The heave and spice processes worked together for the
generation of a strong Pacific cooling region throughout the study period. This co-occurrence of heave and spice processes
intensified severe cooling (Supplementary Figures 5e, f). It is important to note that the sum of the heave and spice appeared
to not be equal to the total trend since there was still a residual term (Zunino et al., 2012). In the Pac sector, the influence of
spiciness became more important during the northward SAF compared to the southward SAF (Figure 9d). There was



significant cooling distributed along all isopycnals from 55° to 62°S (Figure 9d) due to the enhanced cold-water transported

northward and this pattern, like the strong cooling in the total temperature trend in Supplementary Figure 5d, implied that spice dominated the temperature changes in Pac sector. Stronger cooling was also enabled by the tilting isopycnals which then penetrated the interior ocean. Although heave also cooled the same region, the influence was relatively weak compared to that of spice since the displacement of the isopycnals was restricted by the enhanced stratification (Figure 8b). However, the influence of heaving in the Pac sector also played an important role in warming the northern subsurface (Figure 9c). This

subsurface warming was also strong during 1979–2019 (Supplementary Figure 5d) by the subsurface warm water northward. Although heave contributes to warming, it failed to reverse the Pacific cooling which was dominated by the spice process when northward cold-water advection increases during the northward SAF. Overall, spice dominated cooling in the Pac sector during northward cold-water.

## 4 Summary and discussion

Climate change has led to significant warming in the subpolar Southern Ocean, resulting in a distinct pattern of opposing changes in heat due to inverse wind anomalies between the Atl–Ind, near-east Antarctic, and Pac sectors around west Antarctica. The wind field, over a strong meridional temperature gradient in the Southern Ocean, facilitates regional shift of subtropical gyre and brings significant meridional heat advections that capture heat exchange across the subpolar and work as the linkage between mid- and high-latitude ocean changes. In this study, MHA anomalies could be further described as

three terms, $-v'\frac{\partial T}{\partial y}, -v\frac{\partial T'}{\partial y}, -v'\frac{\partial T'}{\partial y}$. The three terms indicate the heat advection due to ocean current variability, warming deficit between subtropical-subpolar basins, and temperature difference caused by turbulence under the circumstance of climatic mean state changes. In the Atl–Ind sector, the first term was caused by the velocity fluctuation due to the southward current, dominated widespread warming at almost all depths, and facilitated surface cooling via sea ice melting. This cooling was confined to the surface (depth < 100 m) and was negligible in preventing overall warming. By contrast, the influence of

all three terms due to the enhanced equatorward flow over the Pac sector was important. The northward cold-water advection, together with the tilting isopycnals, caused remarkable Pacific cooling. This pattern of opposing changes was also reflected in water mass property changes and was properly described by heave and spice processes (Figure 9). A schematic diagram was used to summarize the influence by wind on the opposing changes in the subpolar OHC (Figure 10).





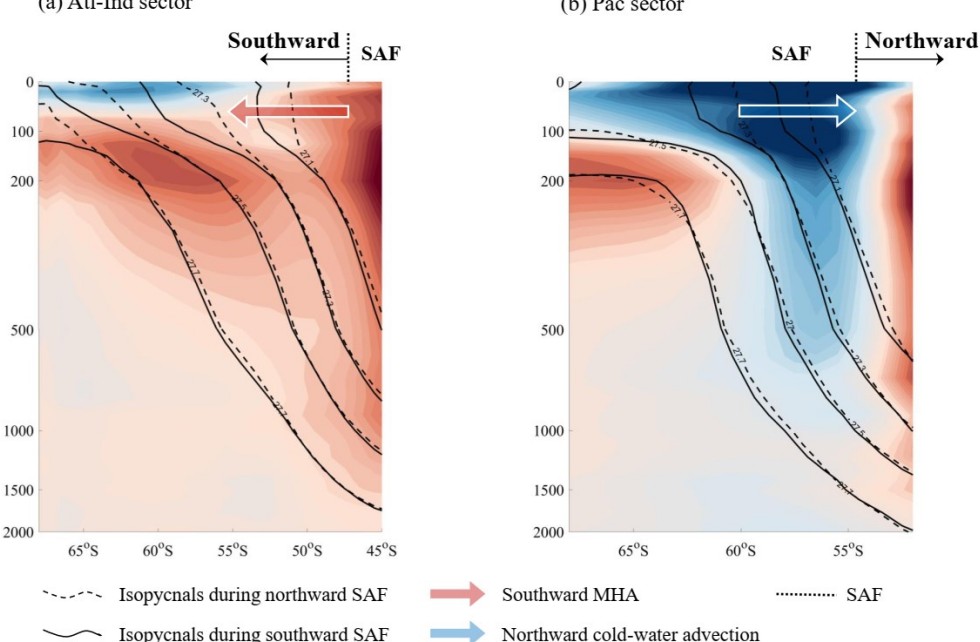

**Figure 10: Schematic of the subpolar OHC changes in the different SAF shifting regions. a) The temperature changes by the southward SAF in the Atl–Ind sector. The shadow from Figure 2c. The dotted lines denote the SAF with the vector to show its shift and red arrows denote the southward MHA. b) same as a) but in the Pac sector. The shadow from Figure 2d and the blue arrows denote the northward cold-water advection. The solid and dashed lines present the isopycnals during the southward and northward SAF, respectively.**

Here, southward SAF and northward SAF denote the poleward heat from subtropical gyre and the equatorward cold-water from Antarctic, respectively (Figure 10). Heave facilitates the displacement of isopycnals and was enhanced during the southward SAF, promoting more heat uptake at the ventilated region and stronger warming of UCDW. The warming in the Atl–Ind sector also had increased downward heat flux on the surface due to less sea ice cover and the water mass changes from the warmer CDW. The significant subsurface warming was also associated with stronger upwelling along isopycnals within the CDW. The Atl–Ind sector was still mainly dominated by significant heave-controlled warming. Moreover, heave may dramatically intensify in the future due to stronger subtropical gyre shifts due to wind and associated significant heat advection, even though shallow cooling resulting from cool Antarctic surface water combined with the cold water by melted sea ice was significant in surface layers. By comparison, the cooling influence of spice processes played a more important role in the Pac sector during the northward SAF. This significant cooling existed in the vigorously ventilated region due to the combination of heave and spice processes. The abundant cold surface water from sufficient sea ice melting was facilitated by the equatorward advection via the stronger negative wind curl and penetrated the deeper layers along the isopycnals. This cooling may be maintained, and even reinforced, as long as there is enough melting to produce cold water and robust ventilation. In this study, we revealed the linkages of the heat exchange, between mid-latitude and high-latitude and detailed that meridional heat/cold-water advection, due to gyre shifting driven by wind curl anomalies, played an



important role in modulating the regional OHC changes, which reflected significant subpolar warming in the Southern Ocean. These linkages are crucial in modulating the heat exchange between Antarctic and subtropical oceans, and will be more important in the warming future.

*Author contribution.* LD provided the initial scientific idea and financial support. LD and XB conceived the idea together for the present study. XB and HY collected all available datasets. XB processed the data, plotted the results, and wrote the first 550 versions of the manuscript. All authors reviewed and edited the paper until its final version.

*Competing interests.* The authors has declared that none of the authors has any competing interests.

### Acknowledgments

This work is supported by the National Natural Science Foundation of China (Grants 42230405, 41825012, and 41576020) and the National Key Research and Development Program of China (2018YFA0605701). We also thank to the CSIRO 555 Marine Research MATLAB Seawater Software Library. We also thank LetPub (www.letpub.com) for its linguistic assistance during the preparation of this manuscript.

### Data Availability Statement

WOA 18 dataset and WOD 18 profiles are available and provided by the National centers for environmental information (NCEI) in (https://www.nodc.noaa.gov/OC5/woa18/;https://www.ncei.noaa.gov/products/world-ocean-database). MOAA 560 GPV observations is provided by JAMSTEC (https://www.jamstec.go.jp/e/). WOCE sections are free provided by CCHDO and are openly accessible online (https://cchdo.ucsd.edu/). The Ishii dataset are available at https://climate.mri-jma.go.jp/pub/ocean/ts/v7.3/ and SODA is released from University of Department of Atmospheric and Oceanic Science, University of Maryland (https://www2.atmos.umd.edu/~ocean/index.htm). ERA5 dataset is available by ECMWF and the Copernicus Programme and the ERA5 dataset has been obtained in CDS (https://cds.climate.copernicus.eu).

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
