# Peer review of "Opposing changes in subpolar ocean heat content due to meridional heat advection driven by the Southern Ocean wind anomaly"

_EGUsphere, 2024_

## Referee Comment (RC1)

**Review of: Opposing changes in subpolar ocean heat content due to meridional heat advection driven by the Southern Ocean wind anomaly**

The authors examined regional changes in Ocean Heat Content (OHC) over two periods, pre-1993 and post-2001. They discovered a notable difference in the patterns between the Atlantic-Indian region and the Pacific region. The Atlantic-Indian region exhibited warming due to increased meridional heat advection induced by poleward westerly winds, while the Pacific region experienced significant cooling driven by equatorward advection caused by stronger winds. The role of the Subantarctic Front (SAF) in influencing heat advection is particularly interesting in explaining  part of these findings.

The paper is well-structured and presents significant findings. Although the authors attempt to interpret the results in light of existing research, this area could be enhanced. The results have potential for publication, but the interpretation and discussion sections need substantial revision to meet the expected standards. I recommend accepting the paper, provided it undergoes major revisions.

My main concern is that the choice of the periods analyzed isn't clearly explained, raising questions about how this might affect the results. The first period has fewer data points, so it's crucial to carefully interpret those findings. While the paper's "Data and Methods" section introduces several oceanographic datasets, it doesn't clearly specify which dataset is used in each analysis within the results section. The figure legends require particular attention to accurately describe each panel. The "Summary and Discussion" section is comprehensive, but it should include more references to earlier studies, which are absent in the current manuscript version. I also suggest a thorough review of the manuscript to refine the writing and improve clarity.

Specific points:

Title:
- I recommended either removing the word "Opposing" from the title or adding the word "Regional." This adjustment can help to clarify the title and make it more accurate or specific in terms of its scope.
Abstract:
- Lines 11 – 13: "Our study reveals that the heat exchange between Antarctic and subtropical oceans driven by wind, which plays an important role in modulating changes in regional ocean heat content (OHC) through meridional heat advections."  Should be rephrase.

- Lines 13 -14 "In this study, we used the observed objective analysis and reanalysis datasets to explore the changes in subpolar ocean heat content and analyze attributions to the remarkable regional discrepancy": I suggest that you mention that you're analyzing the heat content by region, and then describe the significant discrepancies in the following sentence.

Introduction:

- Lines 62 – 70: Need references.

- Line 78: "Importantly, the pattern of subpolar OHC changes exhibits a remarkable discrepancy, but the fundamental factors influencing this discrepancy are unclear.": You should describe the 'discrepancy'.

Data and Methodology:
- Why did you choose to use all these oceanographic datasets? Throughout the manuscript, it is unclear which datasets your analyses are based on (with the exception of the first two figures).
- Why are you using these two specific periods (pre-1993 and 2001-2019)? The trends appear consistent throughout the entire timeframe. How reliable is the data from the first period? There is much more data available for the second period. What impact does this discrepancy have on your analysis? I believe these points should be clarified in the methodology section and discussed in your final section.

Results:

-Line 196: It is significant for all datasets?

- Table 1: The table is not clear, should be re-organize and better explain in the legend. Your top line indicates that both columns 1 and 2 is about OHC and I understand that is just for the first one. I suggested clear state that the first column is OHC of Upper 2000m, AAIW and UCDW, and second column is SAF shift position and MHA.

- Line 230: I only see a change of 0.2°C. Perhaps you should adjust the figure axes to increase the range or visibility of the data.

- Line 231 – 238: Please specify which region you're referring to. It seems like you're discussing the Pacific Sector, but this isn't explicitly stated. Also, are the changes you're observing statistically significant? This should be made clear.

- line 372 – 372: How does this relate to the subtropical gyre below?

- lines 374: I would change 'one and a half times larger' to 50%.

- line 378: In the text, you should explain what RCC stands for.

- line 389: 'Instead of saying "What's more," I suggest using "Additionally."

- lines 390 – 392: It is a confuse statement.

- lines 391-398: You should better demonstrate the influence of the Amundsen Sea Low (ASL), or provide references that support your statement.

- The "Summary and Discussion" section aligns with the results, but there's a lack of comparison with previous studies.

Figures:
- All figure legends need improvement. In some cases, there are panels without a legend at all.

Figure 1:
- Why is there only one significance area for all datasets? It would be more appropriate to have a separate significance area for each dataset.
- Why the colorbar of the panel 'c' is different from 'a' and 'b':
- Enlarge figures a, b, and c so that the label "MOAA GPV" fits within the continent boundaries.
- In panel 'd', please specify in the legend what the colors represent and the meaning of the correlation index.

Figure 2:
- It's difficult to understand what's happening with salinity in panels 'c' and 'd'. I suggest adding two separate panels to isolate the temperature and salinity analyses. You could also include contours indicating where the trend is significant for each parameter.

Figure 4:
- The legend for panel 'a' needs more detailed explanation, similar to how you did it in panel 'b'.

Figure 6:
- Instead of calling panel 'a' a time series, consider naming it "anomalies and differences." Additionally, panel 'b' lacks a legend; please add one.

Figure 8"
- The legend for panel 'd' is missing.

Figure 9:
- There's a slash ('/') in the color bar title that shouldn't be there. Consider removing it.

---

## Author Comment (AC3)

**Response to anonymous reviewer#2 for manuscript "Opposing changes in subpolar ocean heat content due to meridional heat advection driven by the Southern Ocean wind anomaly" by Ni et al.**
**May 2024**

Dear Reviewer,

Thank you very much for your illuminating and constructive comments. Your crucial comments help us a lot to improve this study. This document includes our replies to all your questions. All revises are in blue and green text in the manuscript. If these modifications are still insufficient, please point out the shortcomings, we will continue to improve. Thank you again for your time.

Best regards,

All Authors

The author comment is presented in the following sequence: (1) itemized comments from the reviewer in black, (2) author's response in blue, (3) quotations from the revised paper in *green italic*.

**[Major concerns]**

1.  " They claimed that they "found a notable difference in OHC trends between the Atlantic-Indian sector and the Pacific sector". However, the "Pacific sector" is defined as a region that covers only about a quarter of the real Pacific sector. If the whole Pacific sector is used, I don't think the OHC is decreasing. In any case, I agree that the OHC increase is not uniform in the Southern Ocean, but it is definitely not OK to manipulate the definition of basin sectors."

    Thank you for this crucial comment. We rewrite all the expression to "southwest Pacific sector" (swPac) and revise our description in new manuscript and figures.

2.  " Most of the explanations for how surface wind affect OHC changes are speculative without quantification. To explain the OHC changes, a budget analysis is required -- zonal heat advection or vertical heat fluxes, including air-sea fluxes, may be equally/more important, and different component may compensate for each other. The decomposition into isopycnal heaving and spicing changes is interesting, but the regional isopycnal movement is not necessarily related to meridional heat advections -- zonal convergences may also contribute to or even dominate isopycnal movements."

    Thank you for this crucial comment. About the OHC budget, we refer to the heat budget analysis given by Tamsitt et.al (2016) who replied this heat budget analysis on the temperature trend over the upper 624 m in the Southern Ocean. Here, the heat fluxes, meridional heat advection, zonal heat advection, and vertical convection make main contributions to heat changes. They are calculated as followed:

    $$MH = \underbrace{-\int \rho c_p u \frac{\partial T}{\partial x} dz}_{meridional\ advection}\ ,\ ZH = \underbrace{-\int \rho c_p v \frac{\partial T}{\partial y} dz}_{zonal\ advection},\ VH = \underbrace{-\int \rho c_p w \frac{\partial T}{\partial z} dz}_{vertical\ convection}$$

    where MH, ZH and VH denote the depth-integrated meridional heat advection, the depth-integrated zonal heat advection, and the depth-integrated vertical convection, respectively. Net heat flux through air-sea surface (Q) is given by ORAS5 dataset. Q, MH, ZH and VH are regional-integrated (unit: J/s or W). The results of OHC budget also support the result that meridional heat advection dominates the regional opposite OHC changes between the Atlantic−Indian sector and southwest

Pacific sector. The OHC heat budget is given in section 2.2 Method and we revise and add these investigations as section 3.4 in new manuscript. On the other hand, we add the discussion about the contributions of zonal convergences to isopycnal movements in section 4.
(These add contents are as followed in blue italic)

*section 2.2 Method:*

*"Additionally, regional changes of OHC are closely related to zonal heat advection, net heat fluxes of the air–sea surface, and vertical convection as well. To explain the OHC changes in detail, a budget analysis is required. we refer to the heat budget analysis (Tamsitt, 2016) shown by equation (6) and give the OHC budget (unit: J/s) equation in this study, calculated as equation (7):*

$$\underbrace{\frac{\partial OHC}{\partial t}}_{OHC\ trend} = \underbrace{\frac{Q(z)}{\rho c_p dz}}_{net\ heat\ flux} - \underbrace{\boldsymbol{V} \cdot \nabla_H T}_{horizontal\ advection} - \underbrace{w \cdot \frac{\partial T}{\partial z}}_{vertical\ advection} + diffusion \tag{6}$$

$$\underbrace{\frac{\partial \int \rho c_p T dz}{\partial t}}_{OHC\ trend} = \underbrace{Q}_{net\ heat\ flux} - \underbrace{\int \rho c_p u \frac{\partial T}{\partial x} dz}_{meridional\ advection} - \underbrace{\int \rho c_p v \frac{\partial T}{\partial y} dz}_{zonal\ advection} - \underbrace{\int \rho c_p w \frac{\partial T}{\partial z} dz}_{vertical\ advection} + residual \tag{7}$$

*where OHC is calculated by equation (1), but the unit is converted into J/s (W). The terms on the right of equation (6) are the net heat flux (Q), horizontal advection (AH), vertical convection (VH) and diffusion), respectively (Tamsitt, 2016). In this study, the horizontal advection (unit: W) includes meridional advections (MH) and zonal advections (ZH) and they are calculated by equation (7). Q is the net air–sea heat flux (unit: W). The vertical advection (VH) is related to subductions, convections, as well as the meridional overturning circulation etc. The diffusion term is relatively small then other factors in Tamsitt (2016) but this term is important in our study since when considering the upper-2000m OHC trend rather than temperature trend of thinner layer, the diffusion is involved in the residual term which also includes other important influence such as the active eddy activity in the Southern Ocean (Dufour et al., 2015). "*

*section 3.4 OHC budget:*

*" Given complexity of the regional OHC changes, the net air-sea flux, meridional advection, and zonal advection as well as vertical advection were considered (Supplementary Figure 9). Here, all terms in equation (7) were vertical integral and regional integral (unit: PW, 1 PW= $10^{15}$W). The influence by meridional advection was similar to MHA since MH was the vertical integral and regional integral of MHA. In the Atl–Ind sector, ocean mainly gained heat in the southwest Indian sector ($0^o$–$60^o$E) during period 2 compared to period 1 (Supplementary Figure 9a). This large heat uptake occurring in the Indian sector after 1990s was also mentioned by Song (2020). At the same time, meridional advection presented an obvious banded distribution along longitude (Supplementary Figure 9b) while zonal advection changed relatively sightly (Supplementary Figure 9c). ZH changes exhibited a small patchy distribution since the zonal advection was associated with the recent changes of strong zonal current. The zonally heterogeneous accelerations of ocean current induced the heterogeneous zonal advections and also contributed to the heterogeneous heat changes (Shi et al., 2021).*

*The vertical advection was closely related to the downward Ekman pumping driven by wind, and also related to the upwelling of warm CDW (Tamsitt, 2016). Downward Ekman pumping facilitated surface heat into ocean and this heat uptake occurred in Ekman layer. The upwelling CDW took heat upward from deep to upper-500 m when it accessed to the continental shelf (Meijers et al., 2011). However, VH changed little between period 1 ad period 2. Vertical advection only showed increased in the north of the swPac sector since there was significant subduction of SAMW which brought plentiful heat into ocean (Liu and Huang, 2012). The residual changed significantly compared to VH and ZH since there was turbulent diffusion as well as the influence by active eddy activities (Tamsitt, 2016; Dufour et al., 2015). Strong eddy activities in the Southern Ocean brought heat from the surface to interior ocean potentially and make deep ocean warming. Additionally, influence by residual also involved the topographic, which resulted in thinning of water column in the upper ocean and consequently the heat content reduced. In the swPac sector, the meridional advection decreased (Supplementary Figure 9b) and the residual increased that meant they played a opposite role in regional OHC changes in the swPac sector, and also indicated that the meridional advection and residual term were the main contributor to regional OHC changes. Then, the OHC budget in each sector was given to quantify the contributions of each factor (Figure 10).*

*OHC budget showed the importance of each factor (Figure 10). The OHC trend was multiplied by ten to show clearly since it was relatively small mainly due to the influence of meridional advection as well as the residual. Regional OHC trend consisted of net air-sea heat flux, meridional advection, zonal advection, vertical advection, and the residual. In the Atl–Ind sector, the increasing of subpolar OHC was dominated by the meridional advection, which is also consistent with Wei (2023). MH increased from 19.1 PW in period 1 to 27.2 PW in period 2. But the residual, followed by zonal advection and net heat flux, still tried to restrict the heat content increasing. Although the adverse impact from the residual and from zonal advection increased by 67.5% and 18.8%, the OHC trends still increased 0.96 PW from pre-1993 to post-2005 (Figure 10a). Southern Ocean has absorbed a great deal of heat but ocean still loss heat in region sea surface since the ocean was relatively warmer than cold air. Thus, Q presented a negative effect. The vertical term was small since that the large vertical temperature gradient usually occurred within thin layer near surface and ocean vertical velocity was also small generally. Therefore, meridional heat advection played a dominated promotive role in increasing the OHC in the Atl–Ind sector during the whole study period. And the OHC trend was the results of the competition between MH, ZH and the residual.*

*In the swPac sector, MH was also the most important contributor though the contributions to OHC budget in the swPac was half of those in the Atl–Ind sector. The negative OHC trend in period 2 was three times than that trend in period 1 (Figure 10b). Meridional cold-water advection was dominated and MH changed from –6.5 PW in period 1to –7.6 PW in period 2. The negative net heat flux was also conducive to decreasing OHC though its working was weak. The residual also increased by 15.1% from pre-1993 to post-2005. The residual had a negative effect comparing to MH on decreasing OHC. The oppositive effects of residual and MH were also significant in their regional distribution (Supplementary Figure 9b, e). In comparison to ZH and VH in the Atl–Ind sector, zonal advection and vertical advection were both small that were also shown in Supplementary Figure 9c, d. Hence, meridional cold-water advection played a dominated*

*promotive role in decreasing the OHC in the swPac sector during the whole study period. And the OHC trend was the results of the competition mainly between MH and the residual. "*

[Figure]

*Supplementary Figure 9: The net air-sea flux (Q), meridional advection (MH), zonal advection (ZH) and vertical advection (VH) difference between period 1 and period 2 (Term$_{2005–2019}$ minus Term$_{1979–1993}$). (a) Q difference, the positive denote ocean gaining heat. (b) MH difference, the positive denote northward heat advection. (c) ZH difference, the positive denote eastward heat advection. (d) VH, the positive denote upward heat advection. (a) residual difference, the positive denote ocean gaining heat.*

[Figure]

*Figure 1: OHC budget. a) OHC budget in Atl–Ind sector. The purple, yellow, and white bar denote terms in period 1, period 2, and whole period (1979–2019), respectively. The clusters from left to right present OHC trend (trend is multiplied by ten to show clearly), net air-sea flux (Q), meridional advection (MH), zonal advection (ZH) and vertical advection (VH). b) same to a) but in swPac sector.*

*section 4 Summary and discussion:*

*"Given the complication of regional OHC variations, a budget was given to present the quantification of net heat flux, horizontal advection, and vertical advection (Figure 10). As mentioned above, meridional heat advection dominated the recent OHC trend, especially after 2005. Similar to Tamsitt (2016), regional mean heat flux mainly occurred at the north of 50ºS, and heat loss was widespread at the south of 50ºS. Thus, in the Atl−Ind sector, the significant poleward MHA was also encouraged by a lot of heat flux absorbed in the subtropical. At the same time, this horizontal meridional accompanied with significant wind curl changes resulted in heat divergence and/or convergence, which also contribute to heave as well as spice processes via heat redistribution (Clément et al., 2022). The residual was upper-2000 m depth integral in the OHC budget and was as important as meridional advection since there were active eddy activities and topographic dissipation, which can induce vertical heat exchange between surface and intermediate ocean as well as horizontal heat transport (Dufour et al., 2015). Additionally, existence of topography may also result in the heat divergence and/or convergence via advection due to thinning of water layers. "*

*References:*

*Clément, L., McDonagh, E.L., Gregory, J.M., Wu, Q., Marzocchi, A., Zika, J.D., and Nurser, A.J.G.: Mechanisms of Ocean Heat Uptake along and across Isopycnals, J. Clim., 35, 4885-4904, doi: 10.1175/JCLI-D-21-0793.1, 2022.*

*Dufour, C.O., Griffies, S.M., de Souza, G.F., Frenger, I., Morrison, A.K., Palter, J.B., Sarmiento, J.L., Galbraith, E.D., Dunne, J.P., Anderson, W.G., and Slater, R.D.: Role of Mesoscale Eddies in Cross-Frontal Transport of Heat and Biogeochemical Tracers in the Southern Ocean, J. Phys. Oceanogr., 45, 3057-3081, doi: 10.1175/JPO-D-14-0240.1, 2015.*

*Liu, L.L. and Huang, R.X.: The Global Subduction/Obduction Rates: Their Interannual and Decadal Variability, J. Clim., 25, 1096-1115, doi: 10.1175/2011JCLI4228.1, 2012.*

*Meijers, A.J.S., Bindoff, N.L. and Rintoul, S.R.: Frontal movements and property fluxes: Contributions to heat and freshwater trends in the Southern Ocean, J. Geophys. Res. Oceans, 116, doi: 10.1029/2010JC006832, 2011.*

*Shi, J., Talley, L.D., Xie, S., Peng, Q., and Liu, W.: Ocean warming and accelerating Southern Ocean zonal flow, Nat. Clim. Chang., 11, 1090-1097, doi: 10.1038/s41558-021-01212-5, 2021.*

*Tamsitt, V., Talley, L.D., Mazloff, M.R., and Cerovečki, I.: Zonal Variations in the Southern Ocean Heat Budget, J. Clim., 29, 6563-6579, doi: 10.1175/JCLI-D-15-0630.1, 2016.*

*Wei, Y.: Roles of thermal advection in setting the Southern Ocean warming pattern, Int J Climatol, 43, 6939-6945, doi: 10.1002/joc.8243, 2023.*

3. " [Figure 2 & discussions] The temperature/salinity differences in density coordinate cannot be compared directly to changes in depth/pressure coordinates. The hydrographical changes in density coordinate represent "spice" changes as defined in Section 2."

Thank you for this constructive comment. Here we not emphasized clearly. Figure gives the opposite changes in density coordinate as well in depth/pressure coordinate which is for exhibiting the opposite changes more specifically. And we described changes of temperature and salinity and always pointed out the specific isopycnal or depth of these changes. About this aspect. we rewrite the descriptions and the sentences before and after it and we also repaint it.

[Figure]

*Figure 2: The regional mean temperature and salinity change based on Ishii. a) The vertical temperature and salinity changes in the Atl–Ind sector in the potential density coordinate. The green (purple) bars represent the temperature difference between the mean temperature of period 1 (period 2) and the climatological temperature of the whole study period (zeros line). This temperature difference is calculated as $T_{period\ 1}$- $T_{climatology}$ and $T_{period\ 2}$- $T_{climatology}$. The green (purple) lines with yellow circles represent salinity difference calculated as temperature difference. The temperature change is on the black x-axis (unit: °C) and the salinity change is on the orange x-axis (unit: psu). The interval of potential density on the y-axis is 0.05 kg/m³. b) same as a) but in swPac sector. c) Zonal mean temperature trend (unit: °C/decade) in the Atl–Ind section. The dotted shows the temperature trend exceeding 95% significant level. d) same as c) but in the swPac sector. e) Zonal mean salinity trend (unit: $10^{-1}$ psu/decade) in the Atl–Ind section. The dotted shows the salinity trend exceeding 95% significant level.    f) same as e) but in the swPac sector.*

*"Note that Figure 2 exhibited the temperature and salinity changes in density coordinate as well in depth/pressure coordinate, which is for exhibiting the opposite changes more specifically. The warming/cooling and salting/freshening were all pointed out the specific isopycnal or depth that the changes occurred at. "*

4. " [Lines 369-398]: I don't see significant wind stress curl changes over the period. I don't understand how these positive wind stress curl that last a few months affect the SAF position? If we consider the SAF as the southern boundary of the subtropical gyre, its position is more likely affected by the long-term mean position of zero wind stress curl. Do you see such changes? Also I don't think Amundsen Sea Low is play a role in the considered region."

Thank you for this helpful comment. We found that these positive curl anomalies occurred more frequently, especially after 1993, which reflected the wind field changes at annual or interannual time scale but not at seasonal time scale. The running correlation between wind curl and MHA also reflected this. We also add the position of zero wind curl changes and repaint figure 6. The position

changes also showed southward shift trend after 2005. Additionally, we revise our description of important process about the influence by Amundsen Sea Low and add reference.

[Figure]

*Figure 6: Wind field changes and correlation between Wind curl and MHA. a) Wind field changes. The arrows denote changes of wind field (2005–2019 minus 1979–1993). b) Wind curl anomalies and differences. The red and blue shadows denote anomalies of the Atl–Ind sector and swPac sector, respectively. The thick red and blue lines present the significant positive significant anomalies and negative significant anomalies, respectively. c) Position of zero wind curl. The thin red and bule lines denote the position of zero wind curl in the Atl–Ind sector and swPac sector, respectively. The red and blue trends denote the movement trend of the position of zero wind curl of the Atl–Ind sector and swPac sector, respectively. Note that the negative trend denotes the position shifts southward. d) Time series of RCC in the Atl–Ind sector. e) same to d) but in the swPac sector. The blank shows the RCC failed to exceed the 95% confidence level from the two-tailed t test.*

*" The position of zero wind curl in the Atl–Ind sector also shifted poleward significantly by –0.04ºS/ decade in period 2 which also manifested the southward subtropical gyre (Figure 6c). The evolution of positive wind curl anomalies was also consistent with the running correlation coefficients (Figure 6d). The change of RCC indicated those periods that wind curl has a significant effect on MHA. In the Atl–Ind sector, positive RCC meant that positive wind curl enhanced and southward heat advection also increased."*

*"But in the swPac sector, advections were associated with negative wind curl anomalies, which was dominated in this sector (Figure 6b). The negative wind curl anomalies also occupied frequently for long periods of time and the occurrence was increased by 50% after 1992 than before. Additionally, the positive anomalies were reduced significantly from pre-1993 to post-2005 and the amplitude of positive wind curl decreased by 85% (Figure 6b) at most. These increasing of the negative wind curl and decreasing of positive wind curl reflected the enhance of the Amundsen Sea Low (ASL). ASL was a powerful atmospheric low-pressure system and dominated the meridional flow over the south Pacific (Jena et al., 2022). When ASL enhanced, there was a cyclonic anomaly existing on the swPac sector, and the south wind strengthened as well (Raphael, 2007; Goyal et.al,*

*2021). Then, stronger south wind carried more cold air northward and decreased the sea surface temperature (SST) and increased more sea ice production (Cerrone and Fusco, 2018). Therefore, the northward cold-water advections increased. "*

*References:*

*"Jena, B., Bajish, C.C., Turner, J., Ravichandran, M., Anilkumar, N., and Kshitija, S.: Record low sea ice extent in the Weddell Sea, Antarctica in April/May 2019 driven by intense and explosive polar cyclones, npj Clim. Atmos. Sci., 5, 19, doi: 10.1038/s41612-022-00243-9, 2022.*
*Raphael, M.N.: The influence of atmospheric zonal wave three on Antarctic sea ice variability, Journal of Geophysical Research: Atmospheres, 112, doi: 10.1029/2006JD007852, 2007.*
*Goyal, R., Jucker, M., Sen Gupta, A., Hendon, H.H., and England, M.H.: Zonal wave 3 pattern in the Southern Hemisphere generated by tropical convection, Nat. Geosci., 14, 732-738, doi: 10.1038/s41561-021-00811-3, 2021.*
*Cerrone, D. and Fusco, G.: Low-Frequency Climate Modes and Antarctic Sea Ice Variations, 1982–2013, J. Clim., 31, 147-175, doi: 10.1175/JCLI-D-17-0184.1, 2018. "*

**[Minor comments]**

1.  " "subpolar": When we talk about the subpolar Southern Ocean, we are usually referring to the Weddell and Ross gyres, not the Southern Ocean between 40S and 60S."
    Thank you for this helpful comment. We rewrite the statement of the "subpolar Southern Ocean" in this study in the Section Methods.

    *"In the present study, the subpolar ocean denotes the sectors of the Southern Ocean between 40°S and 60°S. We employed four ocean temperature and salinity datasets to analyze the heat content characteristics in the subpolar ocean. "*

2.  "[Line 15]: What is an "inverse meridional heat advection"?"
    Thank you for this helpful comment. The "inverse meridional heat advection" means that the northward meridional heat advection in the Atl-Ind sector and southward meridional cold-water advection in the swPac sector have opposite direction and influence. We rewrite it to a more understanding phrase: opposite meridional heat advection.

3.  " [Line 17]: What is "poleward westerly wind"?"
    Thank you for this helpful comment. We rewrite it to a more understanding phrase: the poleward shifting of westerly wind.

4.  "[Figure 1 captions]: (c) the OHC changes should be OHC post 2005 minus OHC pre-1993"
    Thank you for this incisive comment. We had rewritten the Figure 1(c) caption.

    *"c) The OHC changes (unit: $10^{19}$ J) derived from WOA between the period 1 and period 2 (OHC $_{pre-1993}$ minus OHC $_{post-2005}$)."*

5.  " [Lines 183-189]: Why define such an index? Can you just use stratification?"

Thank you for this crucial comment. Stratification gives the vertical changes of isopycnals in specific depths indeed. But the stratification index used in this study can quantify the strength of stratification over 2000 m in each period which reflects the overall influence by the vertical distribution of isopycnals in the upper ocean during difference climate change periods.

6. "[Line 201]: The cooling is more widespread in WOA (Figure 1c). I don't understand this sentence -- what do you want to say by "it still attained -0.1 ZJ/decade"?"

Thank you for this crucial comment. We want to express that the pattern of OHC trend showed that the unexpected cooling occurred but was restricted in the southwest Pac sector, and this decreasing of OHC reached −0.1 ZJ/decade. We rewrite this sentence to express clearly.

*" There was unexpected cooling occurring but restricted in the southwest Pac sector, and this decreasing of OHC reached −0.1 ZJ/decade. "*

7. " [Figure 5]: What do "southward SAF" and "northward SAF" mean here?"

Thank you for this crucial comment. The "southward (northward) SAF" means that regional SAF shifts southward (northward) and we revise this in Figure title.

*" The southward (northward) SAF denote that regional SAF shifts southward (northward)".*

8. " [Line 373-374]: What is a "north positive wind curl"?"

Thank you for this crucial comment. We revise this sentence. The "north positive wind curl" means that the positive wind curl field shifts northward.

9. "[Figure 9]: Do you calculate heaving and spicing in depth coordinate and then map to isopycnal coordinates?"

Thank you for this crucial comment. Here we give the heaving and spicing in depth coordinate (shown as the following figure). The temperature changes related to heave and spice in depth coordinate also supported the results that the significant warming in the Atl–Ind sector was closely associated with heave and spice and cooling in the swPac was mainly reflected in spice process.

[Figure]

*Figure 9: Temperature change by the heave and spice. Original frame and green frame denote the temperature trend in density coordinate and depth coordinate, respectively. a) The temperature*

*trend (unit: °C/decade) related to the heave process during the southward SAF in the Atl–Ind sector. b) same as a) but of the spice process. c), d) same to a), b) but for the swPac sector during the northward SAF. (e)-(h) same to (a)-(d) but in depth coordinate.*

10. "There are a number of grammatical errors in the writing that makes the paper a bit hard to follow. I would also suggest the readers to streamline their manuscript to make the discussion more concise."

Thank you for this helpful comment. We take your comments about writing seriously. We are revising and polishing this manuscript.